# Preferred Place of End-of-Life Care Based on Clinical Scenario: A Cross-Sectional Study of a General Japanese Population

**DOI:** 10.3390/healthcare11030406

**Published:** 2023-01-31

**Authors:** Kyoko Hanari, Sandra Y. Moody, Takehiro Sugiyama, Nanako Tamiya

**Affiliations:** 1Health Services Research and Development Center, University of Tsukuba, Tsukuba City 305-8575, Japan; 2Hinohara Memorial Peace House Hospital, Nakai 259-0151, Japan; 3Department of Post-Graduate Education, Kameda Medical Center, Kamogawa City 296-0041, Japan; 4Department of Medicine, Division of Geriatrics, University of California, San Francisco, CA 94143, USA; 5Department of Health Services Research, Institute of Medicine, University of Tsukuba, Tsukuba 305-8575, Japan; 6Diabetes and Metabolism Information Center, Research Institute, National Center for Global Health and Medicine, Tokyo 162-8655, Japan; 7Institute for Global Health Policy Research, Bureau of International Health Cooperation, National Center for Global Health and Medicine, Tokyo 162-8655, Japan

**Keywords:** advance care planning, end of life, general population, preferred place of care, clinical scenario-based survey

## Abstract

In Japan, which has an aging society with many deaths, it is important that people discuss preferred place for end-of-life care in advance. This study aims to investigate whether the preferred place of end-of-life care differs by the assumed clinical scenario. This clinical scenario-based survey used data from a nationwide survey conducted in Japan in December 2017. Participants aged 20 years and older were randomly selected from the general population. The survey contained questions based on three scenarios: cancer, end-stage heart disease, and dementia. For each scenario, respondents were asked to choose the preferred place of end-of-life care among three options: home, nursing home, and medical facility. Eight hundred eighty-nine individuals participated in this study (effective response rate: 14.8%). The proportions of respondents choosing home, nursing home, and medical facility for the cancer scenario were 49.6%, 10.9%, and 39.5%, respectively; for the end-stage heart disease scenario, 30.5%, 18.9%, and 50.6%; and for the dementia scenario, 15.2%, 54.5%, and 30.3% (*p* < 0.0001, chi-square test). The preferred place of end-of-life care differed by the assumed clinical scenario. In clinical practice, concrete information about diseases and their status should be provided during discussions about preferred place for end-of-life care to reveal people’s preferences more accurately.

## 1. Introduction

The Japanese population is aging [1]; the Japanese Ministry of Health, Labour and Welfare (MHLW) is building a system that can comprehensively provide medical and nursing care in the community [2]. It is expected that the number of deaths in old age will further increase in the future [1]; however, a previous study showed the prevalence of incapacity to consent to treatment or admission was 45% for a psychiatric setting and 34% for a medical setting [3]. Healthcare providers may not be aware of a person’s medical and nursing care preferences and, as a result, may provide unintended medical or nursing care. Therefore, it is essential to promote advance care planning (ACP). ACP is the process of understanding and sharing a person’s preferences regarding future medical treatment and care [4,5]. Its goal is to coordinate preferences for care with the care received [6]. It is especially important for people to discuss, in advance, the place of care during the end of life, because it is believed that dying in a preferred place is an important aspect of a good death for both patients and their families [7]. Despite these advantages, even if some individuals with an illness can hold a discussion with other people, other individuals may not necessarily accept engaging in ACP due to barriers such as prognostic uncertainty, insufficient time during clinical encounters, poor knowledge about ACP, and physician communication skills [8,9,10]. Therefore, it is important to start ACP and think about the preferred place of end-of-life care before people become incapable of making decisions for themselves.

Several studies have shown that the home is the most frequently preferred place of end-of-life care (31–84%) among the general population [11,12,13,14,15,16]. These research results have had a stimulating effect on policies to promote home care. Although about half of people preferred home as a place of care, the rest preferred other places in general. The proportion of deaths at home in Japan was 15.7% in 2020 [17], and one of the reasons for this may be that some people do not want to receive medical or nursing care at home. Thus, it is important to clarify which factors are associated with choosing the preferred place of care. Previous studies have shown that the preferred place of care was associated with age [12,18], sex [11,18], marital status [13], concern about family burden [18], life experiences (education, hospitalization of a relative, perceived ability to get better pain relief, and longer hospitalization) [12], and familiarity with home care in the general population [11]. In these and other previous studies, respondents were not asked about their preferred place of end-of-life care in the context of a specific disease. Therefore, it remains unclear whether the preferred place of end-of-life care differs according to the assumed clinical scenario. We hypothesized that the preferred place of care would differ depending on the clinical scenario, due to the differences in symptoms that can generally occur depending on the disease. If the preferred place of end-of-life care varies by the assumed clinical scenario, it would be important to include concrete scenarios of a disease in discussions with people. If it becomes clear that the preferred place of care differs depending on the assumed disease, it may be necessary to formulate a policy to improve the system for providing medical and nursing care not only at home but also at places other than home. Accordingly, the purpose of this study is to investigate whether the preferred place of end-of-life care differs by the assumed clinical scenario.

## 2. Materials and Methods

### 2.1. Study Design and Participants

In this cross-sectional study, we used data from the Survey of Public Attitude toward Medical Care at the End of Life, a nationwide population-based survey conducted by the MHLW in December 2017. This survey was conducted by the MHLW every five years to investigate interest in and attitude toward the end of life in the general population. The survey was administered to a nationally representative sample of the general population in Japan aged 20 years and over. The survey used a stratified two-stage random sampling method. In the first stage, 150 sites were randomly selected from the 2015 census enumeration district so that the number of samples at each site was about 20 to 47. In the second stage, individuals who were mailed surveys were randomly selected from the basic resident registers of each of the 150 sites. The survey was mailed to 6000 individuals along with a return envelope and a letter from the MHLW explaining the study, followed by reminders to nonrespondents a maximum of two times.

All the survey data obtained from the MHLW were anonymized. Consent was assumed if the participant chose to complete the survey. The study protocol was approved by the Institutional Review Board of the University of Tsukuba (approval no. 1254, date 30 October 2017).

### 2.2. Measures

#### 2.2.1. Dependent Variable

Our dependent variable was the preferred place of end-of-life care chosen from among home, nursing home, or medical facility. We used questions that contained disease scenarios to investigate whether the preferred place of end-of-life care differed by the assumed clinical scenario.

#### 2.2.2. Main Predictors

The main predictors were the assumed clinical scenario in which the respondents were asked the following questions based on three disease scenarios. Scenario 1 (cancer) stated, “Where would be your preferred place to receive care if you had terminal cancer, could not eat well, felt short of breath but had no pain, and your consciousness is alert and [you were] able to make decisions?” Scenario 2 (end-stage heart disease) stated, “Where would be your preferred place to receive care if you had end-stage heart disease and needed support eating meals, changing your clothes, and toileting, but your consciousness was alert and [you were] able to make decisions?” Scenario 3 (dementia) stated, “Where would be your preferred place to receive care if you had terminal dementia and had difficulty knowing where you were and recognizing your family members, and needed support eating meals, changing clothes, and toileting?” The words “terminal” or “end-stage” were included in each question to mean that the condition was incurable, and that the prognosis was less than one year. For each scenario, respondents selected one of three options: home, nursing home (i.e., long-term care facility), or medical facility (i.e., hospital). In Japan, a nursing home is a facility that provides 24-h operational support and cares for persons who require assistance with activities of daily living and who often have complex health needs and increased vulnerability [19]. A medical facility includes hospitals that provide medical care, such as medical wards, surgical wards, intensive care units, palliative care units, and nursing wards intended for home discharge preparation.

#### 2.2.3. Potential Associated Variables

Demographic characteristics (age, sex, living arrangements, and academic attainment), presence or absence of a family doctor, experience caring for a loved one, the experience of the death of a loved one, and discussion of future treatment were included as potentially associated variables. These variables were chosen because on previous studies [11,20,21]; preferred place of care was associated with age, marital status, education level, end-of-life preparations, and experiences related to death and dying. Age was categorized as decades, starting with ages 20–29 and ending with age 80 or over. The response categories for education level were “elementary school”, “high school”, “junior college”, “bachelor’s degree”, and “graduate school”. The survey did not ask respondents about their household income or other health conditions.

### 2.3. Statistical Analysis

Descriptive statistics examined baseline characteristics, including demographic characteristics (age, sex, living arrangements, and academic attainment), presence or absence of a family doctor, experience caring for a loved one, the experience of the death of a loved one, and discussion of future treatment.

Next, we illustrated the proportion of the preferred place of end-of-life care for each scenario. We examined whether the distribution of preferred place of end-of-life care was uniform using the chi-square test with four degrees of freedom. We then investigated the effect modification of the association between the assumed clinical scenario and the preferred place of end-of-life care by sex and age (<65 and ≥65 years), based on the definition by the Japan Geriatrics Society [22] and the Health Care Insurance System [23]). We tested effect modification using the likelihood-ratio test and the Wald test. For the likelihood-ratio test, we constructed (i) a multinomial logistic regression model to investigate the association of preferred place of care and hypothetical disease and sex or age and (ii) a multinomial logistic regression model with a product term of hypothetical disease and sex or age to address the effect modification, which was followed by the likelihood-ratio test between models (i) and (ii). We considered clustering answers by individuals using generalized estimating equations for the Wald test. Then, we performed the Wald test to examine the homogeneity of the proportional distribution by age or sex.

All statistics were performed using Stata^®^ 15 (StataCorp, College Station, TX, USA). The level of statistical significance was set at *p* < 0.05.

## 3. Results

There were 973 respondents. Listwise deletion was used to exclude responses with missing values of age, sex, and preferred place of care. Data for 889 respondents, equivalent to 91.4%, were analyzed, giving an effective response rate of 14.8% on the national survey from which our data were drawn. Details of the selection process are shown in Figure 1.

Table 1 shows the characteristics of the respondents. Less than half (53.1%) were aged 60 and older, 56.0% were men, and most (81.8%) lived with at least one family member. Nearly 55% had at least a junior college education. Less than half had a primary physician (40.9%), 37.4% experienced caring for a loved one, 41.7% experienced the death of a loved one, and 39.5% discussed future treatment if they had a severe illness.

The preferred place of care in the assumed clinical scenario is shown in Figure 2. Preferred place of end-of-life care differed by the clinical scenario, because the distributions were statistically significantly different for each clinical scenario (*p* < 0.0001, chi-square test). For the assumed clinical scenario of cancer, most (49.6%) respondents chose home as a preferred place to receive end-of-life care compared with a nursing home (10.9%) or a medical facility (39.5%). For end-stage heart disease, a medical facility was most often chosen (50.6%). For dementia, a nursing home was commonly chosen (54.5%). Additional analyzes also showed that the preferred places of care actually selected were deviated from expectations in each disease (Appendix A).

The preferred place of care by the assumed clinical scenario stratified by age or sex is shown in Figure 3 and Figure 4, respectively. For age, the proportion who chose home in the assumed clinical scenario of cancer by age <65 and ≥65 was 53.8% vs. 44.3%, medical facility by the scenario of end-stage heart disease was 51.2% vs. 49.9%, and nursing home by the scenario of dementia was 64.8% vs. 41.8%. There was an effect modification by age (likelihood-ratio test, *p* < 0.001; Wald test, *p* < 0.001) but not by sex (likelihood-ratio test, *p* = 0.21; Wald test, *p* = 0.06) for the association between the assumed clinical scenario and the preferred place of end-of-life care. When the relationship between the preferred place of care and the assumed clinical scenario of dementia was examined by age, those <65 were significantly more likely to choose a nursing home and not home compared to a medical facility, but when end-stage heart disease was examined, those <65 were not likely to choose home compared to a medical facility.

## 4. Discussion

In this study, we investigated the differences between the preferred place of end-of-life care according to three assumed clinical scenarios among the general population in Japan. Respondents chose their preferred place of end-of-life care based on the three scenarios. For the assumed clinical scenario of cancer, the largest percentage of respondents chose their home as a preferred place to receive care at the end of life. However, medical facilities were most often chosen if respondents had end-stage heart disease, and nursing homes were chosen in the dementia scenario. As these distributions showed statistically significant differences, the results suggest preferred place of end-of-life care differs by the clinical scenario. In addition, our findings suggest that the relationship between the preferred place of care and the assumed clinical scenario is modified by age. Previous population-based studies that investigated preferred places of end-of-life care did not include the assumed clinical scenario as a variable. In this setting, a home was most often chosen as the preferred place of end-of-life care [11,14,15,18]. To our knowledge, this is the first study to show that the preferred place of end-of-life care differs by the assumed clinical scenario. The results suggest that providing concrete information about a disease during discussions about a preferred place of end-of-life care more accurately reveals people’s preferences. In addition, these results are important because they may suggest that it is necessary to consider policies to promote the enrichment of end-of-life care, not only at home, but also at places other than home.

Potential explanations for the association between the assumed clinical scenario and the preferred place of end-of-life care include the following. First, we believe the difference in preferred place of care based on the assumed clinical scenario might be associated with knowledge about and images of each disease included in the scenario. For the assumed clinical scenario of cancer, the largest percentage of respondents chose home as a preferred place of end-of-life care, which is consistent with previous studies [18,24]. In Japan, when people think of end-stage cancer, they may imagine that end-of-life care is best provided in their homes, surrounded by family members. This is likely true, because previous studies that investigated components which might contribute to a good death in cancer care in Japan showed that spending enough time with one’s family, having family at one’s bedside, living in calm circumstances, and living in a home-like environment may contribute to a good death [20].

In contrast, when people think about end-stage heart disease, they assume that care should be provided in a medical facility because people believe that those with end-stage heart failure need intensive symptom management, including palliative care [25,26,27]. It is very likely that the general population might imagine that this care can only be provided in an inpatient setting. When dementia was considered, the largest percentage of respondents chose a nursing home as a preferred place of end-of-life care. The trajectory of dementia, in contrast to other diseases, is gradual and accompanied by a progressive decline in cognition and function [28,29]. When people think about end-stage dementia in Japan, they may believe that having end-stage dementia would be a burden on their family [30], and therefore would prefer nursing home placement to lessen the potential burden.

Other studies came to a different conclusion. Among those investigating the preferred place of care for patients [18,31,32], one study showed that patient preferences regarding place of care varied among their diagnoses [33]. As previous studies have shown, public opinion often does not coincide with the views of individuals who are close to death [34]. Moreover, the patient’s health status may change over time, leading to changes in their preferences for care [35,36]. Accordingly, our findings have a few implications for clinical practice. When discussing the preferred place of end-of-life care in the future, including the assumed clinical scenario in the discussion may help a person better visualize their preference for the place of end-of-life care. It is certainly challenging to present all clinical scenarios when the preferred place of care is discussed with others. However, healthcare providers should consider raising the subject of common diseases that occur in the general population, such as dementia, cancer, and chronic heart or pulmonary disease, with different trajectories.

Second, we believe that the structure and funding of the healthcare system in Japan may influence people’s preferences for the place of end-of-life care. In contrast to the United States, where hospice care is provided to all under the federal Medicare insurance program based on a life expectancy of fewer than six months irrespective of disease, in Japan, only patients with cancer or HIV diagnoses can be admitted to a palliative care unit. The United Kingdom, Germany, and Canada have similar systems to that of the United States. In addition, in Japan, when people are 65 and older, home care is provided under the Long-term Care Insurance (LTCI) system regardless of disease status; however, for those aged 40–64 years who need home care, the LTCI system will only cover 16 diseases (end-stage cancer and dementia are included). Knowledge of the LTCI system may influence people’s choices of preferred places for end-of-life care. Therefore, healthcare providers in Japan should be aware of the possibility that patient knowledge about the healthcare system may have an impact on their choices for the preferred place of care. In addition, in countries where palliative care is available to all regardless of disease status, healthcare providers should be aware that the preferred place of care may differ based on the type of disease.

Last, our findings showed that the relationship between the preferred place of end-of-life care and the assumed clinical scenario is modified by age. When the relationship between the preferred place of care and the assumed clinical scenario of dementia was examined by age, those <65 were significantly more likely to choose a nursing home, and not home, compared to a medical facility, but when end-stage heart disease was examined, those <65 were not likely to choose home compared to a medical facility. Previous studies from Japan [11,12,18] indicated that those who were less than 65 years of age chose home as a place of end-of-life care compared to a medical facility; however, in those studies, the place was not selected after assuming the disease. The preferred place of care in the context of the assumed clinical scenario can change with age, and thus, it would be important to discuss the preferred place of end-of-life care repeatedly over time.

Our results did not show a significant difference in the preferred place of end-of-life care and the assumed clinical scenario based on sex, which was found in previous studies [11,18]. Interestingly, in the Wald test, which clustered answers by respondents, we found a smaller *p*-value with this test than in the likelihood-ratio test (no clustering), which indicates that each respondent’s answers for the different scenarios were “dispersed” across the diseases, i.e., “different” rather than concentrated on one choice.

### Limitations and Strengths

This study has several limitations. First, the study had a low response rate and was conducted in one country, making it difficult to generalize the results. In addition, we cannot rule out the possibility that the proportion of the preferred place of end-of-life care for each scenario would change due to selection bias, with those who are not interested in end-of-life care probably not responding to the survey. Although studies with high response rates are needed, low response rates and selection bias, as shown in Figure 2, do not seem to invalidate the results of this study. A second limitation is that the present study sampled the general population rather than patients with specific diseases; thus, our results may not be generalizable to a patient population. It is important to ask about preference for a place of end-of-life care to support the hopes of a person; however, the preferred place of end-of-life care does not necessarily correspond to better care, because each place has its limitations in terms of medical treatment provided or care received. Additionally, a previous study indicated that differences exist between pragmatic and ideal end-of-life preferences [35]. In actual discussions of the preferred place of end-of-life care with patients, healthcare providers need to consider the best place based on the patient’s preferences as well as the status of the disease, symptomatology, and family situation. Third, we are unable to reveal whether the respondents chose the place of care according to the physical conditions or the specific disease in each clinical scenario. Additionally, we were unable to ascertain the reasons respondents chose a specific place of end-of-life care because questions addressing this issue were not included. Therefore, there is a need for surveys that include options that standardize physical conditions and differ only in naming the disease and studies that provide detailed reasons why respondents choose different places of end-of-life care. Finally, our study did not investigate effect modifications beyond age and sex. Previous studies have shown that marital status [13], concern about family burden [18], life experiences (education, hospitalization of a relative, perceived ability to get better pain relief, longer hospitalization) [12], and familiarity with home care [11] may influence people’s choices of the preferred place of end-of-life care. Further studies evaluating these possible relationships may help us improve how the preferred place of end-of-life care is discussed with the general population.

The major strength of our study is that the same respondents selected each preferred place of end-of-life care while considering the assumed clinical scenario, which allowed us to examine how the assumed clinical scenario might influence people’s choices. When the preferred place of care is discussed, providing concrete information about the disease or disease status would help people decide or better think about where they are at the end of life. Awareness of our findings on the part of policymakers could lead to policies that promote enhanced end-of-life care, not only at home but also at places other than at home.

## 5. Conclusions

The preferred place of end-of-life care differs by the assumed clinical scenario. More studies are needed to further elucidate the reasons for these choices. In clinical practice, providing concrete information about a patient’s disease during discussions about a preferred place of end-of-life care is recommended for more accurately revealing a person’s preferences. Additionally, regular discussions about a preferred place of end-of-life care may be important due to an effect modification by age for the association between the assumed clinical scenario and the preferred place of care. When formulating policies regarding the preferred place of end-of-life care among the general population, it is necessary to proceed with discussions based on an understanding that not only those who choose to remain in their own homes but also their preferred place may change depending on the assumed clinical scenario. Awareness of the findings here on the part of policymakers can lead to policies that promote enhanced end-of-life care, not only at home but also other than at home.

## Figures and Tables

**Figure 1 healthcare-11-00406-f001:**
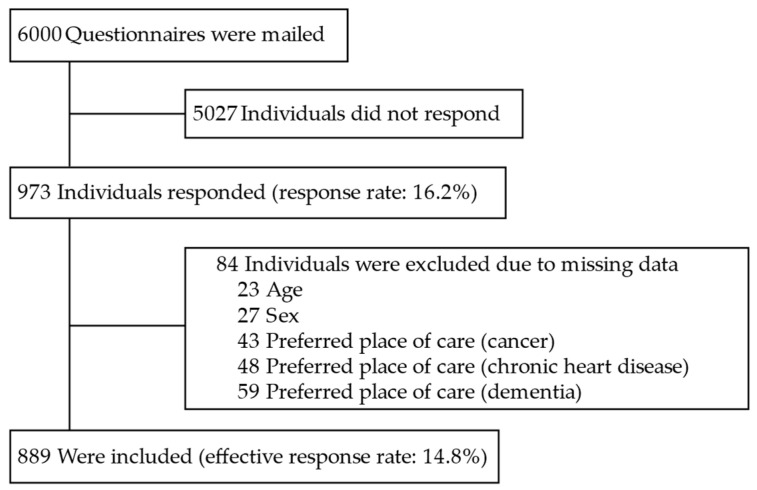
Flowchart of respondents. The 84 individuals missed at least one of the variables; the total number of missingness, therefore, exceeded 84.

**Figure 2 healthcare-11-00406-f002:**
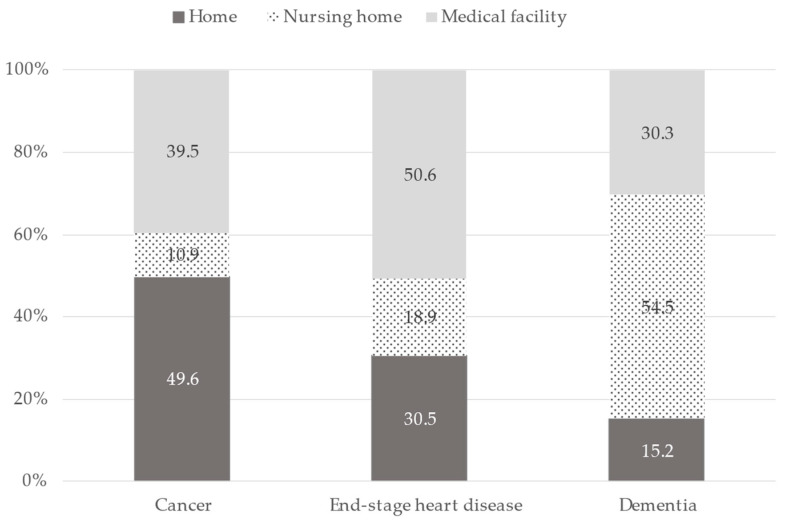
Proportion of preferred place of care by assumed clinical scenario, N = 889.

**Figure 3 healthcare-11-00406-f003:**
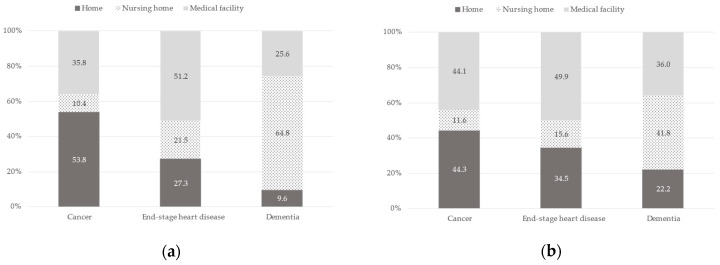
Proportion of preferred place of care by assumed clinical scenario stratified by age: (**a**) Age < 65 (*n* = 492); (**b**) Age ≥ 65 (*n* = 397). Likelihood-ratio test: *p* < 0.001, Wald test: *p* < 0.001.

**Figure 4 healthcare-11-00406-f004:**
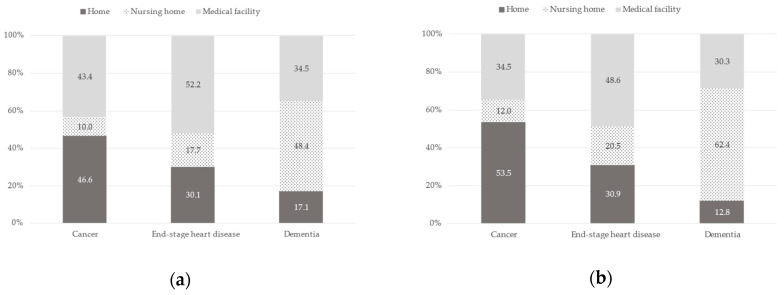
Proportion of preferred place of care by assumed clinical scenario stratified by sex: (**a**) Men (*n* = 498); (**b**) Women (*n* = 391). Likelihood-ratio test: *p* = 0.21, Wald test: *p* = 0.06.

**Table 1 healthcare-11-00406-t001:** Characteristics of the respondents, N = 889.

Characteristics	N (%)
Age, years	
<65	492 (55.3)
≥65	397 (44.7)
By decade	
20–29	39 (4.4)
30–39	107 (12.1)
40–49	137 (15.4)
50–59	133 (15.0)
60–69	188 (21.1)
70–79	188 (21.1)
≥80	97 (10.9)
Sex	
Men	498 (56.0)
Women	391 (44.0)
Education	
Elementary school	92 (10.4)
High school	306 (34.4)
Junior college	171 (19.2)
Bachelor’s degree or graduate school	317 (35.7)
No answer	3 (0.3)
Living arrangements	
Alone	142 (16.0)
With ≥1 family member	727 (81.8)
No answer	20 (2.2)
Has a family doctor	
Yes	364 (40.9)
No	518 (58.3)
No answer	7 (0.8)
Experience caring for a loved one	
Yes	332 (37.4)
No	549 (61.7)
No answer	8 (0.9)
Experienced death of a loved one	
Yes	371 (41.7)
No	484 (54.4)
No answer	34 (3.8)
Discussed future treatment	
Yes	351 (39.5)
No	505 (56.8)
No answer	33 (3.7)

## Data Availability

The datasets used and analyzed during the current study are available from the corresponding author upon reasonable request.

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
