# Peer review of "Preferred Place of End-of-Life Care Based on Clinical Scenario: A Cross-Sectional Study of a General Japanese Population"

_healthcare, 2023, doi:10.3390/healthcare11030406_

Round 1

Reviewer 1 Report

Thank you for giving me the opportunity to review this manuscript. The manuscript is well written, but I would like to suggest a few comments for authors consideration.

Reference number 3: decontextualized study, almost 13 years old.

Reference 7: no more recent literature available? This is a well-researched topic

Line 50: point out which barriers are involved

Line 53: in Japan or in general?

Line 74: what questions and with what criteria were proposed to MHLWs?

Line 116: Is the factor having a physician in the family in these studies? Not specified in the explanation of the choice of these variables.

In the flow chart, review the breakdown of excluded participants. Does not agree with the total

there are quite old references that should be updated

Do the authors think that culture could play an important role in the choice of the place where they wish to die? 

Reviewer 2 Report

line 21 - questionnaire is a survey so do not need both words

maybe include recommendation in conclusion

line 41 change one of the 'decisions' in this sentence to another word. Reads better

line46/7 - change one of 'preferred place'

line 49 hold discussions with who

be consistent with use of survey versus questionnaire

line 76 this is not clear and needs explaining more

line 104 were these definitions put into the questionnaire as well

table format with lines went wrong

think it would be useful to include in abstract the difference that age found as it is the question I was asking myself as I read the abstract. The significance of age also need to feature in discussion more - from line 232

line 263 I assume you mean in relation to discussing this with people - needs adding

line 287 should be new paragraph as new point discussed

line 297 adding 'making' into this sentence before 'generalising'

line 300 sentence is not clear, neither is the next sentence

line 304 should be new paragraph and this sentence is not clear

in conclusion need more discussion of the results

add recommendations

Reviewer 3 Report

It is valuable work to investigate the preferred place of end of life care. I think that data was collected in relation to this topic and meaningful results were obtained. However, in order to be published in this journal, this manuscript needs to be supplemented more. These are the things that need to be supplemented:

1. Although the scenario in the title is omitted as a research method could be omitted, it is better for the predictors to be revealed.

2. The introduction needs to be supplemented a lot. Most importantly, the introduction should present the rationale that places are varied depending on the type of disease at the end of life, and that it interacts with gender or age groups.

3. Explaining the research design, the author's contribution does not need to be presented in the text of the manuscript, but there are other sections.

4. When describing a dependent variable, a predictor, or other variable, it is faster for the reader to understand it by specifying the variable accurately and then explaining it.

5. It was approved by the IRB in October 2017, and data was collected in the same year. By the way, why did you submit this manuscript now? There were other variables with main purpose, so it was published and manuscript with this topic was submitted later?

Reviewer 4 Report

Thank you for giving me the opportunity to review the manuscript entitled "Preferred place of end-of-life care: a clinical scenario-based survey  of a general Japanese population" submitted to Healthcare.

An interesting and important topic. The paper deals with end-of-life care of the general Japanese population. The authors investigate that the place of end-of-life care is meaningful in home, nursing home, and medical facility.

Although the topic seems interesting, I recommended major revision for publishing the paper. Please find my main comments below that are supposed to help the author(s) in further developing their manuscript:

Title

The title is not suitable - if the goal is preferred place for end-of-life care, why are general population mentioned and why is placed in the tile?

Abstract

Should be cleared what is the main purpose, what are really objectives, what is the methods used, what findings are, what conclusion is?

See line 18-32

Introduction

Overall structure needs to be revised. The introduction should rather clarify the research gap, research question and goal/contribution of their own study.

See line 37-51

Purpose of the study: Needs to be clarified. 

See line 52-66

Methods

Should be more detailed what study design? What study is setting? How selection the population? Why is general population? How many items are assessed?

* should be added ethical approval in the method section.

See line 70-121.

The "analyses" section needs to be substantially revised according to the most recent data analysis standards.

Findings

The main findings are too limited. What results are? What results mean? What results communicated for? What are positive and negative contributions to the objectives?

See line 145-148

See line 154-158

See line 188-194

See line 198-204

I suggest to the authors to better explain the novelties of the approach used and the main findings with previous literature on this topic.

Discussion

1. Should be discussed with the main findings, why are important? What results contributions to similar or different other scholars?

2. Discussion with social contributions

3. Discussion with practical implications

Conclusion

This section is too limited in explaining what the main findings are?

References

Many texts and the reference lists are incorrect.

1. Should be up-to-date

2. Re-checked all texts and related studies

Writing structure

Should be followed with a logical flow of writing or scientific study.

*English

Many grammatical errors and inconsistent sentence presentation. Should be provided a professional or native speaker in the field to edit is required.

Reviewer 5 Report

---------------------------------------------------------------------

Manuscript ID: healthcare-2159133

Preferred place of end-of-life care: a clinical scenario-based survey of a general Japanese population

---------------------------------------------------------------------

The manuscript entitled “Preferred place of end-of-life care: a clinical scenario-based survey of a general Japanese population” studies the preferred places for end-of-life care, choosing between home, nursing home, and medical facility, assuming three different hypothetical clinical situations. The manuscript is brief and concise. This study presents a cross-sectional design, which includes data from general population. The methods seem sound and the measures were adequate and clearly explained. There are some things in the manuscript that should be clarified before publication. Here are some comments I would like to offer to the authors.

1. Introduction and 4. Discussion

In the Discussion section (p. 7) you propose reasons to explain why the respondents chose their preferred places in a given situation. You even provide previous literature to support this. However, could we not say that these results were to be expected? Did the authors think that something else could have happened? The article does not have a hypothesis section at the end of the introduction (and I think it should).

The chosen situations and preferred places are very specific. Why those and not others? It occurs to me, for example, that a part-time nursing home option could have been added. Or a terminal illness situation that would not bias the response toward one of the three options. It is possible (and this should be discussed in the limitations section) that the wording of the items may bias the response toward one of the response options. Not eating well, no pain, and to be able to make decisions leads to choose home; heart desease sounds as a severe medical condition better attended in a hospital; dementia and constant help in  ADL lead to choose nursing home.

2.3. Statistical Analysis and 3.  Results

You reported the chi-squared test p-value, but not its degrees of freedom. I am assuming a 3 x 3 contingency table (and therefore 4 df), but it is better to report all the details for a better understanding.  

Please, also report and interpret the standardized residuals for each cell of the contingency table. Which are the observed proportions that differ from those expected (as per the null hypothesis)?

In addition, please calculate and interpret some index of effect size. If there are no other similar studies to compare with, at least this would be useful for future studies on this same topic. At the very least, it will be useful to compare the magnitude of the relationship between scenario and preferred place between age groups. In any case, a contextual interpretation of the observed percentages would be appreciated.

I do not follow the reasoning as to why the proportion of respondents who reported home as the preferred place for end-of-life care may be overestimated (p. 8).

Round 2

Reviewer 1 Report

Dear editor and authors, thank you for your efforts in making the suggested changes. Those aspects that detracted from the methodological validity of the text have been changed and it can now be accepted for publication.

Reviewer 4 Report

Thank you for the opportunity to review in the second time. This is a good revised version. The paper is written well and based on sound research. I recommend publication subject to the following acceptance.